# TONGUESWITCHER:
# Fine-Grained Identification of German–English Code-Switching

**Igor Sterner**
Department of Engineering
University of Cambridge
United Kingdom
is473@cam.ac.uk

**Simone Teufel**
Department of Computer Science and Technology
University of Cambridge
United Kingdom
sht25@cam.ac.uk

## Abstract

This paper contributes to German–English code-switching research. We provide the largest corpus of naturally occurring German–English code-switching, where English is included in German text, and two methods for code-switching identification. The first method is rule-based, using wordlists and morphological processing. We use this method to compile a corpus of 25.6M tweets employing German–English code-switching. In our second method, we continue pretraining of a neural language model on this corpus and classify tokens based on embeddings from this language model. Our systems establish SoTA on our new corpus and an existing German–English code-switching benchmark. In particular, we systematically study code-switching for language-ambiguous words which can only be resolved in context, and morphologically mixed words consisting of both English and German morphemes. We distribute both corpora and systems to the research community.

## 1 Introduction

A large proportion of the world's population is multilingual, and that naturally means that a large proportion of the world is code-switching daily, frequently and routinely (Harris and McGhee Nelson, 1992; Grosjean, 2010; Grosjean and Li, 2013). Code-switching occurs when speakers alternate between languages; this can happen at the sentence, word, or even subword level. For many multilingual speakers, code-switching is a natural part of informal language, either as a matter of convenience or possibly because it allows them to express their intended meanings more precisely. Several psycholinguistic and sociolinguistic theories of code-switching exist (Poplack, 1980; Joshi, 1982; Myers-Scotton, 1997; Muysken, 2000; Green and Abutalebi, 2013; Filipović and Hawkins, 2019). The dominant language is called the matrix language, while the subordinate language that is included is

---

**Tweet:** ich glaub ich muss echt ~~rewatchen~~ ~~like i feel so empty~~ was soll ich denn jetzt machen
**Translation:** I think I really have to rewatch it like i feel so empty what should I do now

Figure 1: German–English code-switching

called the embedded language (Joshi, 1982). We refer to any text segment in the embedded language as an island. In the example of code-switching shown in Figure 1, ~~like i feel so empty~~ is an island.

Many NLP systems are currently developed to be capable of handling text from informal contexts. Code-switching places new pressure on these, particularly for applications that require the recognition and precise extraction of meaning from code-switched text, or even the generation of such text. Available NLP tools lag behind in this respect (Aguilar and Solorio, 2020; Doğruöz et al., 2021); in particular large language models perform best when fine-tuned on natural code-switching data (Santy et al., 2021). Our work is aimed towards NLP tools that can better understand and manipulate code-switched language.

We are interested in studying *naturally occurring* code-switching. Social media, where mostly informal conversation take place, is an ideal source of such texts. We study code-switching between English and German, two closely related languages. We encounter many high-frequency words of identical form in both languages, such as "was" in Figure 1, which is a WH-pronoun in German. If the same string "was" appeared in English, it would be the past form of 'to be'. Importantly, the two meanings are entirely unrelated. Such cases constitute an interesting corner case for code-switched text, and are called interlingual homographs (IHs, Dijkstra et al., 1999). A second interesting phenomenon is that, because German is a morphologically rich language, its morphology can act on English mor-

phemes, creating intra-word codeswitching, such as in the past participle $_\text{M}$rewatchen in Figure 1. A third property of the German–English language pair is the high frequency of English loanwords in German. Loanwords are words of foreign origin that have been fully assimilated into the main language. Loanwords and code-switching constitute a grey area in language change: whether something is a loanword or part of an island is a hotly debated topic in linguistics (Deuchar, 2020; Treffers-Daller, 2022). While we do not directly address the loanword distinction in this work, we believe that our theory-neutral methods can contribute to an empirical way of addressing this in the future.

In this paper, we study German–English code-switching with an analytic interest in fine-grained phenomena (e.g. short islands, mixed morphology and interlingual homographs). This introduces new aspects to the automatic study of code-switching. At the same time, we bring scale to the analysis; the TONGUESWITCHER Corpus includes 25.6M German-English code-switching tweets with automatic code-switching identification. We release our corpus and the two code-switching identification methods we developed, one rule-based, one neural[1].

## 2 Related work

Code-switching identification and language identification are closely related tasks, but traditional language identification (LI) tools can only determine which languages are present in a given text, not the precise beginning and end of each island. For instance, the LI tools provided by Chen and Skiena (2014) and Joulin et al. (2016b; 2016a) rely on character-based n-gram models. FastText (Joulin et al., 2016b,a) uses a character-based n-gram method to compare statistical properties of the input text with a pre-compiled frequency profile of each language. It distinguishes 176 languages, including English and German, alongside similar languages such as Luxembourgish and Afrikaans. Polyglot is another such tool, which is able to identify more than one single language per document (Chen and Skiena, 2014). It is built from the CLD2 tool from Riesa and Giuliani (2013), which uses quadgram ranking. Lingua (Stahl, 2023) is a black-box LI tool that also offers code-switching identification for many language pairs, including German–

English. It combines a language modelling approach with hard-coded rules. Its code-switching identification performance has never been experimentally evaluated.

Nguyen et al. (2020; 2021) present rule-based code-switching identification systems for Vietnamese–English and Hindi–English mixed text, which is based on specially-created wordlists for each of these language pairs. All words that appear in both wordlists are manually disambiguated by a human annotator. This is a simple approach to the task that affords the researchers control over their system, as it does not require any training.

Osmelak and Wintner (2023) detect code-switching at a finer-grained level. In their Denglisch system, tagging proceeds at token-level, and the following labels are used: **D** and **E** for German and English tokens respectively; **SD, SE** and **SO** for loanwords imported from German into English, from English imported German, and from other languages. There is also an **O**ther category for unclassifiable items, such as punctuation and emojis, and a **M**ixed category for words of mixed morphology.

Several other code-switching approaches also model mixed morphology. Nguyen and Cornips (2016) perform morphological analysis with the Morphessor tool to address Dutch-Limburgish-English code-switching, and Mager et al. (2019) detect intra-word code-switches in German–Turkish and Spanish–Wixarika text using RNNs. Osmelak and Wintner (2023) use CRFs, a supervised machine learning framework, in combination with manually curated features, such as orthography, n-gram, morphology, function words, frequency, lexical components and wordlists. The training material consists of 950 Reddit comments containing 60K tokens, balanced between English and German. They also use automatically-tagged silver-standard data to the tune of a further 31,500 comments (5 million tokens). In contrast, our solution does not require any human-annotated training material.

For neural code-switching identification, the use of word embeddings from a multilingual language model such as mBERT (Devlin et al., 2019) is one possible approach. mBERT is an encoder-only transformer-based model which embeds each token into a 768-D vector. Santy et al. (2021) found it is best suited when fine-tuned on naturally occurring code-switching material. Nayak and Joshi (2022)

---

[1] Code, models (neural tagger and code-switching language model, both with demos) and corpus are all online.

pretrain and fintune a BERT-based model for code-switching identification in Hinglish. They use an existing tool to collect and automatically label a large corpus of tweets.

When it comes to gold-standard datasets, the majority of code-switching datasets are between (1) languages spoken in India with English (Gupta et al., 2021; Nguyen et al., 2021; Adda-Decker et al., 2008), (2) Mandarin with English (Lyu et al., 2010) and (3) Spanish with English (Mave et al., 2018; Samih et al., 2016). For German–English code-switching other than Osmelak and Wintner (2023), Rijhwani et al. (2017) use a mini-corpus consisting of 99 Twitter tweets, which is not publically available. Our corpus is much larger.

## 3 Corpus construction

The German tweets we use as input were collected at scale by Kratzke (2022, 2023). The Twitter language identification algorithm assigns a probable language at the time of writing of each tweet; Kratzke chose those that were deemed German. This resulted in 149.2M input tweets written between April 2019 to February 2023. We clean the tweets (URLs are replaced with <URL>, emojis, emails, phone numbers and mentions are removed) and run FastText language detection on them, only keeping tweets that are re-assigned the German tag or instead assigned an English language tag. This step eliminates many tweets in Luxembourgish and other languages too similar to German for Twitter's language identifier to catch. 123.7M tweets remain after this step. In contrast, Osmelak and Wintner (2023) filter their input to remove those examples for which Polyglot's prediction is not *both* English and German.

To establish a testset, code-switching annotation was performed by the authors of this paper on 1252 tweets. We used the Prodigy annotation tool (Montani and Honnibal, 2018). To use annotation time efficiently, we wanted to make sure that a good proportion of the cases seen had reasonably high code-switching occurrences. The tweets were therefore processed and pre-filtered by a precursor of TONGUESWITCHER. This system differed from the final version only marginally, e.g. in the ordering of the rules and the quality of the multilingual stemming algorithm. We then random-sampled from two subsets: all input tweets (25%), and those with a high proportion of code-switching (75%). System annotations were not removed before hu-

man annotation. There were no explicit guidelines. Annotators discarded tweets in German dialects such as Swiss German, made sure that German indeed acted as the matrix language, and then marked island start and end points in each surviving tweet. In 63.5% of cases, boundaries were moved; this means that annotators did not simply accept the system's suggestions. Regarding the annotation of loanwords, each annotator followed their own intuition about which words were so common as to be used as loanwords, additionally using a context-sensitive definition of loanwords. To establish consistency of the annotation, we randomly sampled 36 tweets consisting of 1172 tokens, which both annotators labelled. Inter-annotator agreement was measured at $\kappa$=0.68 (N=1172, n=3, k=2; Cohen, 1960). The annotators agreed fully on all tokens in 15 out of the 36 tweets. The distribution of island sizes in the resulting testset is given in Figure 2.

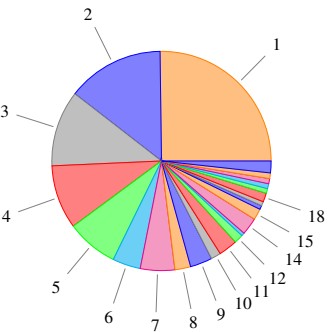

Figure 2: Proportion of island sizes in testset. Each field expresses the total number of tokens occurring in islands of that size.

## 4 TongueSwitcher

Our rule-based method takes as input German tweets and produces labels indicating the language of each word, or sub-word if the word is of mixed morphology. The main algorithm applies several wordlist-based filters to make the decision. Additional processing applies if a) the word is genuinely a possible word in both languages or b) it is an intra-word code switch. All processing in this method is performed on lowercased words.

### 4.1 Constructing wordlists

We first compile formal and informal wordlists for English and German. Our strategy given the resources we have is to compile pure wordlists which contain words that are guaranteed to be contained in only one of the two languages (for instance,

"parser" should not be in either pure list), and a big wordlist to cover as many words as possible from the matrix language, German. For the "Pure English" wordlist, we combine the WortSchatz Leipzig News Corpora (WL, 15K words, Biemann et al., 2007) and a scraped version of the online Urban Dictionary (UD, 13K words, Bierner, 2022), which contains many informal words and phrases used in English slang. We remove words that appear 5 times or fewer in the WL corpus and words that appear 10 times or fewer in the UD.

For German, we use the WL corpora (65K words), and add Swiss (37K words) and Austrian (34K words) as we could not automatically filter out many input tweets written in these dialects. We also add the more informal online German dictionary dict.cc (583K words, Hemetsberger, 2023). This list is the basis for our "Big German" wordlist. For our multilingual stemming, we also need a wordlist of pure German roots. We start by collecting a smaller wordlist of words appearing more than twice only in the German WL corpus.

English loanwords need to be removed from the German lists. Ideally, we would have an exhaustive list of loanwords to handle such words separately, but in reality we have access only to a small list of 3367 known English loanwords in German, created by Seidel (2010) from an analysis of the German magazine Der Spiegel. We remove these from "Big German"[2]. We remove an additional set of suspected loanwords automatically from the German wordlists, namely all entries from dict.cc (Hemetsberger, 2023) where the English word and its German translation are identical and vice versa. We also remove a large list of boys', girls' and city names (Weiss, 2022a,b; OnTheWorldMap, 2023) from both wordlists. Names in our approach are handled based on the surrounding language in our n-gram processing (step **7** of our algorithm coming up in §4.2).

Finally, we also want to remove the many non-language-specific one or two letter words in our wordlists (e.g. "eh"), which we consider noise. Such words can arise from typos, abbreviations, and general processing problems. Unless such ultra-short words were included in hand-selected

| | |
|---|---|
| Big German | 709,979 |
| Pure German | 92,099 |
| Pure English | 20,203 |
| Interlingual homographs | 120 |

Table 1: Wordlists compiled, with number of words

lists[3], we removed them from all wordlists.

Even after all these stages, there are still words appearing in both wordlists (many purely English words in the German wordlists and vice versa). Many of these are noise. One could whittle them down entirely manually, as Nguyen and Bryant (2020) do. We instead first ask the large language model (LLM) `text-davinci-003` (Brown et al., 2020) for its guess of the primary language of each word with the following prompt: `In one word, what language is the word: {}?` The LLM may introduce a bias towards English. We therefore do not accept the LLMs predictions blindly, but manually review all classifications. Knowing the model's choice still saved time. We removed the German ones from the English wordlist, and the English ones from the German wordlists.

When going through this list manually, we also find some words that are graphically identical and have the *same* meaning in both languages (e.g. 'diverse')[4]. If such a word is found, it is removed from *all* wordlists. We will treat these based on the surrounding language later.

Finally, we compiled a list of IHs. Under the assumption that IHs have different POS in the two languages, we compute a list of such IHs by tagging an English and a German WL corpus, looking for shared words with at least one different POS, modulo capitalisation.

The sizes of the resultant wordlists are given in Table 1.

## 4.2 Code-switching identification algorithm

Our code-switching identification algorithm is defined as follows. We first tokenize and POS-tag each cleaned tweet using the Flair `upos-multi` multilingual uPOS tagger (Akbik et al., 2018; Petrov et al., 2012). Then we apply the following steps to classify each token. These steps are also visualized as a flow chart in Figure 3; examples of tokens handled by each step are given in Appendix

---

[2]Another reason for removing these loanwords is that some display mixed morphology. Our algorithm will not detect mixed morphology if the full word is already in "Big German". By removing them, these words are automatically handled by our mixed word detection steps.

[3]The hand-selected lists contain 28 English one or two-character words (e.g. 'of' and 'if') and 24 German one or two-character words (e.g. 'zu' and 'um').

[4]These words are not IHs, because IHs are defined as having different meanings.

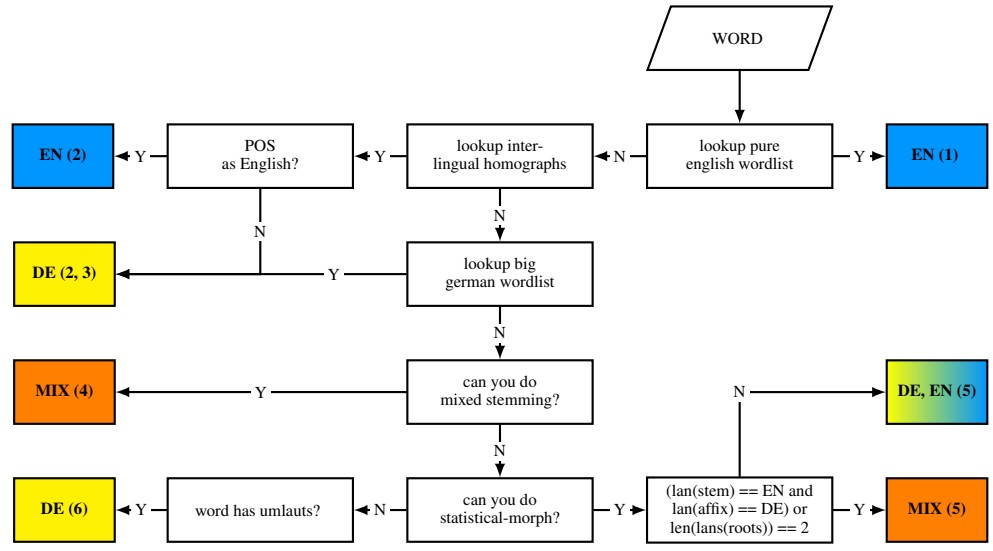

Figure 3: The word-level classification subroutine

Table 9. The proportion of tokens in our corpus identified using each step is given in brackets.

**1** (5.8%) If a word is in our pure English wordlist, it immediately receives an English tag.

**2** (9.3%) If a word is a interlingual homograph (IH), language identification is attempted using the words' part of speech tag.

**3** (74.6%) Else, we look the word up in our big German wordlist and assign a German tag if found.

**4** (0.2%) If the word is still not identified, a multilingual stemming system we developed recursively strips affixes (lists taken from Osmelak and Wintner, 2023) from words until a word (or simple variations: adding a missing trailing 'e' or removing a double last letter) is found in our pure English wordlist or no more affixes are found. If an English stem is found with purely German affixes, the word is given a Mixed label.

**5** (1.4%) We next look for known subwords using a statistical morphological segmentation subsystem based on HanTa (Wartena, 2019), a trainable second-order autoregressive model, where each morpheme depends on the previous two morphemes to predict the most likely morpheme sequence. We train HanTa on the (a) Tiger Corpus (Brants et al., 2002) for German (b) Brown Corpus (Francis and Kucera, 1964) for English and (c) a mixture of both, and attempt segmentation in turn with these three systems, looking for roots in our pure English or pure German wordlists. This subsystem also detects fully monolingual compound word creations, hence the increase in proportion compared to 0.2% for the previous step.

| Split | Tweets | Sentences | Tokens | Eng. tokens |
|-------|--------|-----------|--------|-------------|
| Train | 24.6M  | 57.8M     | 741.9M | 82.6M       |
| Dev   | 1.1M   | 2.6M      | 32.9M  | 3.6M        |
| Test  | 1.3K   | 3.0K      | 37.5K  | 2.8K        |
| Total | 25.6M  | 60.4M     | 774.8M | 86.2M       |

Table 2: TONGUESWITCHER Corpus statistics

**6** (0.5%) If the unknown word contains an umlaut, it receives a German tag.

**7** (8.2%) Words unknown at this stage which occur inside single-language islands are assigned the language of their neighbours. Words at an island boundary assume the language of the most probable bigram on either side, based on the frequencies of the most likely 10,000 bigrams of German and English we compiled from the WL Corpora for each language. Otherwise, tokens assume the language of their nearest identified token.

We implement this algorithm using the framework of Lin and Byrne (2022), resulting in the TONGUESWITCHER (TS) system. Using this system, we next automatically labelled all 123.7M cleaned input tweets, creating our silver-annotated data. Based on the silver labels, we excluded tweets that do not contain at least 50% German tokens, and at least one English or Mixed token. We split this corpus by allocating the last two months of data (Jan, Feb '23) to a development set. A summary of the corpus with its silver-standard training/development data is given in Table 2. Our silver-standard data has 11% English tokens.

We sanity-check our corpus and the silver-

standard annotations by sampling 5 tweets each for different island sizes (1 to 20 tokens). Out of the 100 tweets, 64 were true code-switching. Of the remaining, 9 were translations and 27 were monolingual. Most of the monolingual mistakes arose due to the erroneous identification of single-token loanwords or named entities[5]. Overall, we find the precision acceptable, given that the recall is likely to be higher than in any currently available corpus. If we had chosen a stricter condition for tweets that are selected for our corpus, perhaps to exclude tweets with only a single-token island as Nayak and Joshi (2022) do, we could easily raise precision, but would have missed many interesting border cases of either code-switching insertions or loanwords. It is precisely these instances that are valuable to linguists and lexicographers who study the process of loanwording.

Of the code-switching tweets, we then counted how many of the system-predicted islands were correctly identified[6]. We found that single-token English islands had a precision of 62.5 (40 predicted), two-token islands 87.5 (8 predicted), three-token islands 80.0 (10 predicted) and all island sizes greater than this (65 predicted) had a precision of 100.0.

Two examples of particularly dense code-switching are shown in Table 3. TS labels example (1) perfectly, but for example (2) it incorrectly tags 'performed', 'pushen', 'Time-to-Market' and 'Relaunch' as German[7]. 'Top-of-mind-Awareness' is not segmented correctly by Flair and hence incorrectly identified as the language of the surrounding tokens, which is German.

## 5 BERT-based system

We also wanted a neural system that is fine-tuned for German–English identification, so we could investigate to which degree neural word embeddings are suited to the task. To that end, we pretrain a neural language model on the TONGUESWITCHER Corpus and fine-tune it for token classification. We then learn the classification layer using the human-labelled examples from the Denglisch Corpus. This system is called tsBERT.

---

[5]The TS system does not include any named entity recognizer, or special handling of loanwords, except using the wordlists and surrounding language.

[6]We used a lenient definition of boundaries where overlap between system-predicted islands and real islands was sufficient.

[7]It did so because these words all happen to make their way into our "Big German" wordlist, and are also not in our "Pure English" wordlist.'

## 6 Experiment

**Systems, Competitors, Baselines** We evaluate our systems, TONGUESWITCHER (TS) and tsBERT, against two competitors from the literature, Denglisch CRF (Osmelak and Wintner, 2023) and Lingua (Stahl, 2023). The Denglisch system is not provided as a trained system, so we follow their procedure in training it. To interpret Denglisch's output, we match Denglisch labels onto our reduced set as follows: English, German, Mixed are taken directly. SE becomes English, and SD becomes German. Denglisch's SO labels and punctuation labels are ignored in evaluation.

We construct a strong baseline by prompting the GPT-4 LLM (OpenAI, 2023) with the prompt given in Appendix §A.3. We also train baseline neural classification models by learning the classification layer directly on (English) BERT (e*BERT*), German BERT (g*BERT*, DeepsetAI, 2019) and multilingual BERT (m*BERT*) models.

**Datasets** We use the TONGUESWITCHER Corpus as pretraining data, and the human-labelled examples from the Denglisch Corpus (Osmelak and Wintner, 2023) as finetuning data (after removing emojis, replacing out-of-vocabulary punctuation tokens, and removing entries longer than 100 tokens).

Our main evaluation uses our own corpus (§3) with its 1252 tweet testset. We also report results for our systems and the Denglisch CRF system on the German–English subpart of the Denglisch Corpus (15% of their corpus sentences, using the same definition as before). While our BERT-based system is trained on their data in the cross-validation setup, TONGUESWITCHER cannot be trained. We use this evaluation as a sanity check: if our systems performed much below the Denglisch system on this corpus, this would be a cause for alarm.

**Training** We initialize our BERT-based models with the `bert-base-multilingual-cased` (mBERT) pretrained model (Devlin et al., 2019). Unlike our rule-based system, this model distinguishes between upper and lowercase words. We continue pretraining for 1 epoch on all 24.6M code-switching tweets in the TS training corpus. We finetunne for our task on the Denglisch Corpus (Osmelak and Wintner, 2023). For evaluation on their corpus, we train models for the same 10-fold cross-validation setup as they do. For evaluation on our testset, we train on 100% of their corpus,

| | | | |
|---|---|---|---|
| **(1)** Pronouns: he/him Height: 1,83m Zodiac: Virgo Smoke: nope Tattoo: 3 Piercings: Ohrringe (mehr will ich auch nicht, allerhöchstens noch mehr Ohrlöcher) Fav colour: grün Fav drink: Kaffee und oolong milk tea, heiß, mit einem quarter süss und tapioka bei meinem bubble tea laden | | | |
| **Translation:** Pronouns: he/him Height: 1.83m Zodiac: Virgo Smoke: nope Tattoo: 3 piercings: earrings (I don't want more, at most more ear piercings) Favorite colour: green Favorite drink: coffee and oolong milk tea, hot, with a quarter sugar and tapioca at my bubble tea shop | | | |
| **(2)** Wenn wir unsere Skills elevaten und die Units gemeinsam performen, werden wir die Sales auf ein neues Level pushen. Außerdem können wir so den Time-to-Market für den Relaunch shorten. Das bringt zusätzliche Top-of-mind-Awareness und pushed die Brand in der Community. Ok? Go! | | | |
| **Translation:** If we elevate our skills and perform the units together, we will push sales to a new level. This also allows us to shorten the time-to-market for the relaunch. This brings additional top-of-mind-awareness and pushes the brand in the community. Ok? Go! | | | |

Table 3: Examples from our TONGUESWITCHER Corpus sanity check

as they do when labelling their silver-standard material. Training details are given in the Appendix §A.2.

**Metrics** We report results separately in token-based micro-averaged $F_1$ measure (shown as $F_t$), and in entity-based $F_1$ measure (shown as $F_e$). $F_e$ is defined based on the number of islands of English inside the German matrix text, with strict boundaries. We use the BIO format (Ramshaw and Marcus, 1995) for entity representation. Because code-switching segments are coherent entities inside a text, using an entity-based metric should be more informative than a token-based one, which ignores the code-switching context of each token. We report performance on all islands, and we also introduce a new metric which measures the performance of systems for short islands only, namely those consisting of 2-4 tokens according to our gold standard. The statistical test we use throughout this paper is the two-tailed paired permutation test, approximated by $R = 10,000$, with significance threshold at $\alpha = 0.05$.

## 7 Results

| | **German** | **English** | **Mixed** | **Overall** |
|---|---|---|---|---|
| | *9907* | *1972* | *192* | *12071* |
| *Denglisch* | 97.5 | 89.1 | 25.6 | 95.5 |
| TS | 96.9 | 87.7 | 32.4 | 94.5 |
| *tsBERT* | 98.9 | 95.5 | 60.1 | 97.8 |

Table 4: Results on Denglisch corpus; in $F_t$

Table 4 gives results in $F_t$ on the G–E subset of the Denglisch corpus. Our trained tsBERT model outperforms trained Denglisch in all categories (differences significant; 4x p<0.01), setting a new SoTA on this benchmark. The superiority of tsBERT in the English category (95.5 vs. 89.1), which is the core of the task of German–English code-switching

identification, is particularly satisfying. In mixed word detection, our system achieves a 135% improvement over Denglisch.

Revisiting example (2) from Table 3, where TONGUESWITCHER (TS) made multiple mistakes, tsBERT fixes all of these mistakes and perfectly identifies the code-switching. Denglisch predicts 'Skills', 'elevaten', 'Units', 'performen', 'Relaunch', 'shorten', 'pushed' are all German, and wrongly suggests that 'Time-to-Market' is mixed.

Meanwhile, TS is not trained on any Denglisch data, as it is rule-based[8]. In the English and Mixed categories, TS is statistically indistinguishable from Denglisch (p=0.19, p=0.09); in the German category, it is significantly outperformed by Denglisch.

We consider both our systems to pass the sanity check; we will now turn to our main results on our own corpus, where no new human-annotated training material is available to any of the systems.

Table 5 shows the results in precision, recall and $F_t$ for our corpus.

TONGUESWITCHER ($F_t$=97.1 overall) and tsBERT ($F_t$=97.0 overall) are indistinguishable from each other, and significantly better than all baselines and competitors, with the exception of the category Mixed. In the mixed category, TS is better than tsBERT (p<0.01), and tsBERT is indistinguishable from all BERT-based baselines. All other differences are significant, which means that GPT-4 ($F_t$=94.3 overall) is inferior to our two TS systems, as least with our prompting strategy. This means that TS has established SoTA on our corpus.

TS outperforms all others in the mixed category; the BERT-based models are the next best. Although

---

[8]Note that our treatment of Denglisch's gold standard (collapsing all 'Shared German' tokens to be German) hurts only TS. For example, TS would say named entities like 'Berlin' are English in an otherwise English constituent.

| | German 29761 | | | English 2757 | | | Mixed 129 | | | Overall 32647 | | |
|---|---|---|---|---|---|---|---|---|---|---|---|---|
| | P | R | $F_t$ | P | R | $F_t$ | P | R | $F_t$ | P | R | $F_t$ |
| *Lingua* | 95.8 | 97.3 | 96.5 | 66.5 | 57.6 | 61.7 | 0.0 | 0.0 | 0.0 | 93.6 | 93.6 | 93.6 |
| *GPT-4* | 99.2 | 95.2 | 97.2 | 66.2 | 93.7 | 77.5 | 12.2 | 16.3 | 14.0 | 94.8 | 94.8 | 94.8 |
| *Denglisch CRF* | 98.4 | 97.4 | 97.9 | 75.1 | 85.5 | 79.9 | 19.0 | 6.2 | 9.4 | 96.0 | 96.0 | 96.0 |
| *eBERT* | 98.7 | 97.4 | 98.0 | 78.1 | 86.7 | 82.2 | 23.1 | 38.0 | 28.7 | 96.3 | 96.3 | 96.3 |
| *gBERT* | 98.8 | 97.0 | 97.9 | 73.9 | 87.6 | 80.1 | 27.7 | 34.1 | 30.6 | 95.9 | 95.9 | 95.9 |
| *mBERT* | 98.7 | 97.5 | 98.1 | 78.1 | 87.3 | 82.4 | 24.9 | 32.6 | 28.2 | 96.4 | 96.4 | 96.4 |
| TONGUESWITCHER | 99.3 | 97.6 | 98.4 | 79.0 | 93.8 | 85.8 | 48.0 | 38.0 | 42.4 | 97.1 | 97.1 | 97.1 |
| *tsBERT* | 99.0 | 97.9 | 98.5 | 81.5 | 89.1 | 85.1 | 25.5 | 38.8 | 30.8 | 97.0 | 97.0 | 97.0 |

Table 5: Results on our testset

| | Island 1192 | | | Short Island (2-4) 365 | | |
|---|---|---|---|---|---|---|
| | P | R | $F_e$ | P | R | $F_e$ |
| *Lingua* | 25.4 | 14.0 | 18.1 | 27.8 | 34.5 | 30.8 |
| *GPT-4* | 44.5 | 70.1 | 54.4 | 50.7 | 74.5 | 60.4 |
| *Denglisch* | 49.0 | 55.5 | 52.0 | 53.2 | 72.3 | 61.3 |
| *eBERT* | 54.0 | 61.5 | 57.5 | 63.1 | 70.7 | 66.7 |
| *gBERT* | 49.2 | 58.4 | 53.4 | 55.3 | 71.0 | 62.2 |
| *mBERT* | 54.8 | 62.0 | 58.2 | 63.4 | 73.7 | 68.2 |
| *TS* | 58.9 | 75.7 | 66.2 | 57.3 | 77.3 | 65.8 |
| *tsBERT* | 60.5 | 66.5 | 63.4 | 66.7 | 75.9 | 71.0 |

Table 6: Island-based results

mixed word identification might be seen as a niche task given the low occurrence frequency of mixed words, we are happy to see this result because we think that the mixing of morphologies is an understudied phenomenon. Linguists and cognitive scientists requiring empirical data can profit from a system such as ours that is able to automatically detect these cases reasonably well.

It is nice to see the small, but significant improvement of our tsBERT system over the other BERT-based models in most categories (those other than Mixed). This shows that pretraining with the TONGUESWITCHER code-switching corpus helps. This language model trained on code-switching data may be useful to other researchers working on German–English tasks other than ours, too.

### 7.1 Islands

So far, we have presented results in a token-based metric, but this ignores the fact that code-switching is a context-sensitive phenomenon: we care less about how many tokens are of which language overall, and more about which textual material forms an island.

Table 6 gives results for P, R and $F_e$ for islands and short islands. Again, the two TS systems beat all competitors and baselines. Lingua is left far behind. The success of TS on islands is a surprising result, as the majority of tokens are handled by this system without any context. One explanation may be that step **7** in our algorithm (§4.2) performs contextual smoothing by assigning the labels of neighbouring tokens to unknown tokens. This handles spelling mistakes and other word creations by favouring coherent islands.

For short islands, tsBERT and mBERT are joint winners with $F_e$=71.0 and $F_e$=68.2, respectively, beating TS (65.8), GPT-4 (60.4) and Denglisch (61.3). TS is better than GPT-4 (p=0.02), while GPT-4 and Denglisch are indistinguishable. Lingua's performance, meanwhile, is poor at $F_e$=30.8. We suspect Polyglot (Chen and Skiena, 2014) would have similar problems with this task[9]. Denglisch (Osmelak and Wintner, 2023) use Polyglot as a filtering tool for all their data and therefore many cases of short islands of code-switching may have been lost when the Denglisch corpus was created.

### 7.2 Post-analysis: interlingual homographs

We next performed an analysis of how well the systems perform on IHs. We compiled a separate small testset specifically for such cases: tweets containing real IHs. We sorted our previous list of IHs by the frequency of the less frequent language of the two (e.g. English 'war' rather than the German verb), and then manually checked up to 100 tweets in each language for each word. We discarded words if they show any of the following problems: the word was only ever encountered as a proper name in one or both languages (e.g. English "los"),

---

[9]We base this on the assertion by Lingua's authors that Lingua beat Polyglot experimentally (see GitHub). We have not verified Polyglot's performance; it was unsuitable as a baseline for us, as it cannot predict token-level labels.

| | German | English | Overall |
|---|---|---|---|
| | *146* | *130* | *276* |
| *Lingua* | 70.9 | 55.7 | 64.9 |
| *GPT-4* | 92.8 | 92.0 | 92.4 |
| *Denglisch* | 74.9 | 72.0 | 73.6 |
| *eBERT* | 80.0 | 80.3 | 84.1 |
| *gBERT* | 85.4 | 81.5 | 83.7 |
| *mBRET* | 84.4 | 83.7 | 84.1 |
| TS | 84.5 | 82.8 | 83.7 |
| *tsBERT* | 89.3 | 89.1 | 88.8 |

Table 7: IH disambiguation results (in $F_t$)

---

**(3) Tweet:** I don't get IH was er damit erreichen will.
**Translation:** I don't get what he wants to achieve with that.

---

**(4) Tweet:** fang über nächstes jahr mit abi an but no problem zeugnis durchschnitt IH was 1.5 letztes halb jahr wird schlechter sein dieses halbjahr tho cuz mental health yk
**Translation:** no problem the average result of the end-of-year report was 1.5 last half-year will be worse this half-year tho cuz [because] mental health yk [you know]

---

Table 8: Examples from our IH testset

or the word was so infrequent in German–English code-switching in the target sense that it didn't occur in the top 100 tweets (e.g. English "stark"). We found 29 true IHs with at least one tweet of true English and German usage[10]. For each IH, 2-10 tweets were added to the testset. We attempted to balance the tweets between German and English occurrences and prioritised examples where the IH was at a borderline of an island. This resulted in a testset of 253 tweets with 276 IH tokens, 47% of which were in English.

Results are given in Table 7. For IHs, our rule-based TS (overall $F_t$=83.7) and neural tsBERT (overall $F_t$=88.8) outperform trained Denglisch (overall $F_t$=73.6) and Lingua (overall $F_t$=64.9; all Denglisch and Lingua results significantly different from all other systems). For the BERT systems, in all categories, eBERT is indistinguishable from mBERT, which is indistinguishable from gBERT. TS is indistinguishable from eBERT, gBERT, and mBERT in all categories. Overall and for German tokens, it is also indistinguishable from tsBERT. GPT-4 and tsBERT are indistinguishable in all categories (p=0.21, 0.13, 0.08).

The "strong baseline" GPT-4 and our neural system tsBERT turn out to be best at the hard task

---

[10]Namely *war, bin, bad, see, die, man, was, made, ran, toll, falls, hat, dick, drum, links, still, these, fast, hell, handy, fort, positives, tag, sage, seen, lose, rum, will, not*

of disambiguating these words. Table 8 gives two examples for the IH 'was'. In German, this string is a WH-pronoun, whilst in English it is the past form of 'to be'. All systems except Lingua correctly identify (3) as German. In contrast, the only system to identify the IH in (4) as English is GPT-4.

## 8 Conclusion

We have presented two methods for German–English code-switching identification. Our rule-based system enabled us to collect the largest corpus of naturally occurring code-switching. Our BERT-based model, trained on this corpus and fine-tuned on human-annotated data, established SoTA on an existing German–English benchmark. We also established SoTA on our newly formed corpus using token and entity-based metrics. A post-analysis on interlingual homographs revealed that neural language models are the best systems for disambiguating these words. Overall, our study combines two aspects we think are important for the future of code-switching: a) the use of large-scale empirical methods on naturally occurring data and b) an analytic interest in fine-grained linguistic phenomena.

## 9 Future work

We are interested in providing a more objective definition of loanwording, as opposed to genuine code-switching, in the light of the debate in linguistics, lexicography and cognitive science. Our future contribution to this topic will centre around the fact that the distinction can only be made *in context*, more specifically in island-context. Therefore, it is useful to employ the best tool for island detection, and we have demonstrated here that our systems for German–English are very effective. Frequency also plays a role; loanwords which can be considered part of German will be far more frequent in German matrix text than any naturally occurring English words in English islands. We release frequency-sorted data of the top 10,000 islands of each island length in the TONGUESWITCHER Corpus. This may serve as a starting point for empirical studies of this challenge.

## Limitations

There may be some bias in our gold standard due to the pre-selection of tweets found by TS. In the future, we plan to create a new gold standard entirely from scratch, even if this requires more annotation

effort and guidelines. Our current definition relies on annotators' intuition too much.

Evaluation of systems such as ours is also difficult, partly because code-switching language identification is subjective. In particular, annotators and NLP systems often introduce English bias (Anastasopoulos and Neubig, 2020; Garrido-Muñoz et al., 2021).

In our rule-based system, we do not implement a named entity recognizer. As such, in our corpus, named entities containing English words are often incorrectly labelled as English.

The quality of the multilingual part of speech tagger, alongside its tokenization, also constrains our method. Tagging all our input tweets with this tagger required intensive GPU computation.

In terms of our mixed identification methods, our TONGUESWITCHER system over-segments words (e.g. verrate), which is a particular problem for misspelt words.

## Ethics Statement

Working with and releasing large corpora of social media posts raises data privacy concerns. We do not collect any personal information about the authors of the tweets. We release our corpus to the research community only.

## Acknowledgements

We thank Alexander Bleistein and Constanze Leeb for their early support of this work. We also thank Louis Cotgrove, Chris Bryant, Li Nguyen and our three reviewers for their insightful comments.

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

## A Appendix

### A.1 Corpus examples by algorithm step

| | |
|---|---|
| **1** | Sagt ein(e) Head(in) `of` `research` ! Researchen Sie mal ein bisschen mehr |
| **2** | habn Match mit einer, bin einfach unfähig, dein a `man` with a `hat` , ehem. Schädling |
| **3** | Freigabeworkflow `für` PR Manager `sind` `das` `nächste` Product Highlight `für` Insta u Twitter. |
| **4** | ich bin grade in einem chat am `shittalken` mit einem äußerst platonischen freund |
| **5** | Der Vorstand traf sich letztes Wochenende zum jährlichen `Arbeitsweekend` , dieses Mal in Thun. |
| **6** | Danke ich hänge die dritte Woche mit einer `Nervenwurzelentzündung` durch; Schmerzen trotz starker Medikation tlw from the hell Physio ist gut |
| **7** | joko `in` diesem fußball fit lebt immer noch rent free `in` my mind wie schön |

Table 9: Example of tokens classified in each step
.

### A.2 Training hyperparameters

We use the masked language modelling objective presented by Devlin et al. (2019). We train using 4 NVIDIA A100 GPUs, for approximately 30 hours per GPU. We use a batch size of 32, which amounted to 191,950 steps. We use a learning rate of 1e-4 with a warmup of 10,000 steps followed by linear decay, $\beta = (0.9, 0.999)$ and weight decay $= 0.01$.

To learn the classification, we train for 3 epochs using a learning rate of 3e-5, batch size of 16 and weight decay $= 0.01$.

### A.3 GPT-4 prompt

```
Sentence: {tweet} Task: Fill in the following
list of words and their labels by identifying
each of the words in the sentence as English
('E'), Mixed ('M') or German ('G'). Punctuation
should be the same language as its surrounding
associated words. Mixed words switch between
English and German within the word. Only use
the tags 'E', 'M' or 'G'. Fill in: {token_1:
'', token_2: '', ...}
```

We found the output JSON was rarely malformed or of a different length to the input tokens, but in those cases where it was we repeated the prompt.