# OpenReview forum: "TongueSwitcher: Fine-Grained Identification of German-English Code-Switching"
_EMNLP/2023/Workshop/CALCS — EMNLP 2023 Workshop CALCS_

### Official Review · Reviewer_vrzY · 2023-09-22
**TongueSwitcher: Fine-Grained Identification of German-English Code-Switching**

**Rating:** 4
**Confidence:** 3

**Review:**

## General comments
This is a nice paper with interesting contributions to the field of German-English codeswitching. The research is well described and contextualized. The dataset proposed, the rule-based system for CS annotation and experiments performed are valuable contributions.

There are some assumptions or missing info that I think are key. Clarifying them could improve the paper:
- The dataset selection relies on a preprocessing step to filter tweets that contain CS. However, little is said about that preprocessing step. The point of the paper and the experiments performed is to evaluate how good TONGUESWITCHER is, and we are doing so testing it on a selection of data that was selected by a precursor of TONGUESWITCHER. I think this may questionable, or at least require more info on the preprocessing step.
- The introduction states the importance of the CS/borrowing distinction, which seem to announce that the issue will be covered throughout the paper (also, the "fine-grained" in the title made me think that). But the problem is ignored during the project: the annotation of the CS/borrowing distinction was left to the annotators intuition (no definition is given on the paper, no IAA was computed) and 1-token islands were basically ignored when comparing results among models, even though 1 token islands amount to 25% of the CS segments on the dataset (Fig 2). If the authors clearly stated from the beginning that they decided to skip the borrowing/CS distinction, it would be less disappointing.
- I found the section on creating the wordlists a bit challenging to follow

## Minor comments

- From the 200 mill original tweets, how many tweets remained after the FastText pre-processing?

- Line 236: The authors mentioned an initial set of tweets being processed by a precursor of TONGUESWITCHER to select tweets that were codeswitching-rich, but no more information is given on that pre-processing step. I miss having some info about the workings of that precursor, because the rest of the paper (dataset construction, model results) depend basically on this preprocessing step. Could the precursor affect somehow to the resulting selection of tweets? Also, if the selection of tweets was made using a preprocesor that mimics TONGUESWITCHER I wonder whether the selection of tweets is biased towards what the precursor was able to identify as CS. That would mean that the type of codeswitched tweets that TONGUESWITCHER had to process were those that the precursor found to be codeswitched-rich, which may make the task easier.

- If I understood correctly, after the precursor was run, 1252 tweets were randomly selected and manually annotated as test set. Then an unknown amount of tweets from the original 200 mill tweets were automatically annotated with TongueSwitcher. Those tweets that contained at least 50% tokens of German and at least one English inclusion (according to TS) were included in the silver standard. The silver standard amounts to 20 million tweets automatically annotated by TS (and not reviewed by any humans). If this is correct, did the authors do any sanity check or analysis on what is inside those 20 million tweets?

- Lines 246-250: the criteria to annotate a word as a loanword or not was the speaker's intuition (so, if I understand correctly, well-established English loanwords into German were annotated as German). I miss having some more info on this, given the blurry line between loanword and CS (as was stated on the introduction of the paper). Were there no guidelines used whatsoever? If none were given, was any IAA metric calculated for loanword annotation? It'd be helpful to assess how reliable the speakers' intuition was. The paper mentions giving a context-sensitive definition of what a loanword is, but that definition is not given.

- More generally, how was the annotation of those 1252 tweets done? Was there any double annotation? Any guidelines? Did they compute IAA?

- Besides loanwords, how did the annotators deal with issues such as named entities in English (for example, if "The New York Times" appeared in a tweet, how was that annotated? Was that considered a CS?). How did they deal with Twitter English lingo such as "lol" or "hashtag" (these are English words frequently used in Twitter conversation that don't imply bilingual discourse).

- Figure 2: perhaps it'd be more expressive if instead of pie graph, an x/y graph was provided, with the number of tokens in the island plotted on axis x and the number of islands plotted on axis y. The current pie graph gives an idea of the proportion, but makes it difficult to compare one section with another and the total number of islands per length is not given.

- Line 284: what were the authors trying to avoid by removing 2-character strings? I assume that they identified some noise, but I'd like to know a bit more about what type of noise it was and why was it 2 character strings precisely (and not 1 character or 3)

- Line 300: I can't avoid being a bit skeptic about the quality of the monolingual wordlists. After all, one of the main points of the paper is dealing with ambiguous words that can exist is both German and English. A LLM was used to filter words from the list, in a word-by-word fashion. Although the output of the prompt was reviewed, I can't help but wonder whether potential ambiguous words skipped from the filtering system (so, for ex, a word that on a shallow look is a German word but that could also function as an English loanword).

- When building the wordlists, after the lang id step with text-davinci-003, I guess that all words in the German wordlists labeled by davinci as English are removed, correct? Similarly, with English, correct? I assume that is the case, but I don't think it's explicitly mentioned on the paper. Also, the words that were processed by davinci were the ones that appear both in the German wordlist and the English wordlist, correct?

- I'm a bit confused by how many wordlists were created. From reading section 4.1 I understood that there were 1 list per language (and then the additional CLH list), but Table 1 lists 3 word lists (none of them being the CLH list).

- If I understood correctly, in addition to the English wordlist and the German wordlist, there was a third list with 120 crosslingual homographs. Correct? This was created "by tagging an English and a German WL corpus, looking for shared words with at least one different POS". Was that annotation done by hand or automatically? Were the words in the CLH list excluded from the English wordlist and the German wordlist? This list is the one that is used in step 2 of the rule based algorithm. I think this step becomes easier to understand if the third list compilation is mentioned at the end of section 4.1.

- Line 376: the authors state for the first time that what they are introducing is a silver-annotated data. While the resource is indeed still very useful, I think that the fact that the resource they are releasing is silver standard (so automatically annotated) should be mentioned earlier on the paper (abstract or introduction)

- Line 456: when the authors say that they exclude single-token island I understand that they simply skip those tokens (so it won't matter whether the goldstandard label on the test set and the predicted labeled matched). This raises 2 comments: (1) If single token islands were going to be ignored to avoid the borrowing/CS distinction, why not remove all sentences with single token islands from the test set to begin with? (2) Given that loanwords can be multitoken (for ex, "fake news", "big data"), there is a chance that among 2-token islands there were in fact 2-token loanwords.

- Also, given that is the most numerous type of island, I would have liked to see metrics for single-token islands. If the authors don't want to go into the loanword/CS debate, they can call those single-token islands as "code-mixed inclusions".

- It's very nice to see they performed statistical test for significance.

- Figure 4: caption should mention that the metric is F1 over tokens

- I miss having some examples of errors and some overview of the types of errors that each type of system made

- I found Section 8 to be somewhat difficult to follow because the summary of what was done was mixed with future work (and mixed with thoughts on the importance of frequency to detect loanwords). The section could improve by being reformulated so the conclusions is clearly separated and the contributions are clearly listed. Many readers will likely scan the abstract and go straight to the conclusions section, so section 8 should be clear and easy to follow on its own.

**Candidate For Best Paper:**

No

**Reason For Best Paper:**

-

**Related:**

5: It is very related to the workshop.

---

### Official Review · Reviewer_MFv2 · 2023-10-03
**Interesting approach to CS identification, but not generalizable to other language pairs**

**Rating:** 3
**Confidence:** 4

**Review:**

**Summary**

In this paper, the authors make 2 primary contributions:
1. Release a corpus of naturally occurring English-German code switched tweets, which consists of a human annotated test set as well
1. Propose 2 methods of identifying code-switching, one rule based called TongueSwitcher (non-generalizable) and one automated (generalizable), that achieve SoTA on their dataset and strong performance on an existing benchmark.

**Pros::**
- The authors run comprehensive experiments across island sizes and also calculate token-level and entity-level F1 scores, which shows the success of their system for both hard and easier cases.
- The authors also present separate experiments for cross-lingual homographs, which in my opinion are one of the hardest problems in code switching.
- Even though TongueTwister is highly specific to the language pairs, the authors ensure comparison with all existing baseline systems on both their dataset as well as existing datasets. This adds a lot of confidence to their claims.

**Cons:**
- the successful TongueSwitcher method presented by the author is rule based and is highly specific to EN-DE language pair and is thus not very generalizable.
- The authors claim that their methods: `require neither high quality lexica nor human annotated training material`. Even though this is true, their ruleset still requires a lot of manual supervision to build the wordlists.
- In section 3, it is not very clear how the authors use a precursor of the Tonguetwister system. That step is unclear to me.
- The quality of TongueTwister seems to be contingent upon the quality of the WordLists, POS Tagger and the segmentation subsystem. It would be interesting to see some ablation studies around them.

**Candidate For Best Paper:**

No

**Reason For Best Paper:**

N/A

**Related:**

5: It is very related to the workshop.

---

### Official Review · Reviewer_nu5X · 2023-10-05
**Good work**

**Rating:** 4
**Confidence:** 4

**Review:**

This paper focuses on English-German code-mixed NLP tasks. They create a large dataset for English-German code-mixing and propose code-switching identification methods. Their systems establish SOTA on existing En-De benchmark. They find that rule based systems excel at detecting mixed words and neural models excel at detecting language ambiguous words (words that have same surface form, but different meaning).


Pros:
- Useful corpus for future research in En-De code-mixing
- Good list of baselines and neural models

Cons:
- Rule based system might be specific to this language pair and can be harder to translate to other pairs.

**Candidate For Best Paper:**

No

**Reason For Best Paper:**

NA

**Related:**

5: It is very related to the workshop.